# Evaluation of new strain (AAD16) of *Beauveria bassiana* recovered from Japanese rhinoceros beetle: Effects on three coleopteran insects

**Souvic Sarker**[1], **Hyong Woo Choi**[2], **Un Taek Lim**[2]*

1 Department of Entomology, Rutgers University, New Brunswick, New Jersey, United States of America,
2 Department of Plant Medicals, Andong National University, Andong, Republic of Korea

* utlim@andong.ac.kr

**Data Availability Statement:** All relevant data are within the paper and its Supporting information files.

## Abstract

A strain (AAD16) of the entomopathogenic fungus *Beauveria bassiana* (Balsamo) Vuillemin was isolated from field-collected Japanese rhinoceros beetle, *Allomyrina dichotoma* (L.) (Coleoptera: Scarabaeidae). Its virulence was compared with another strain (ARP14) recovered from a cadaver of *Riptortus pedestris* (F.) (Hemiptera: Alydidae) focusing on its effect on three coleopteran, i.e., *Tenebrio molitor* L., *A. dichotoma*, and *Monochamus alternatus* Hope. The $LT_{50}$ value of *T. molitor* for two larval sizes, i.e., 16–18 and 22–24 mm, was 15.3 and 19.4% lower for strain AAD16 compared to strain ARP14, respectively. Furthermore, after 8 and 10 days of exposure, the mycosis rate of strain AAD16 was 1.3 and 1.2 times higher than that of strain ARP14 in the 16–18 and 22–24 mm larval sizes, respectively. The $LT_{50}$ for *M. alternatus* larvae was 23.2% lower on strain AAD16 than on strain ARP14. In addition, the $LT_{50}$ for *M. alternatus* adults was 47.1% lower for strain AAD16 compared to control. The mycosis rate of strain AAD16 on *M. alternatus* larvae was 1.8 higher than that of strain ARP14 after 120 hours of exposure. The strain AAD16 also showed higher larval mortality (90%) for *A. dichotoma* compared to strain ARP14 (45.0%) at 28 days after exposure. These results suggest that *B. bassiana* AAD16 can be a potential biological control agent against coleopteran pests.

## Introduction

Biological control is an important component of Integrated Pest Management due to its high efficiency and minimal negative effects on the environment [1, 2]. Predators, parasitoids, and entomopathogens including fungi, bacteria, viruses, and nematodes have potential for use as biological pest control agents [3]. Entomopathogenic fungi (EPF) are effective biocontrol agents of several destructive pest species [4, 5]. Several species of entomopathogenic fungi have been used for pest control, including *Beauveria bassiana* (Balsamo), *Metarhizium anisopliae* (Metschnikoff), *Nomuraea rileyi* (Farlow), *Verticillium lecanii* (Zimmerman), and *Paecilomyces fumosoroseus* (Wize) [6]. Nonetheless, only a small number of entomopathogenic fungal species have been effectively marketed as biopesticides, and there is a need to continue developing better species. Among these species, *B. bassiana* is a candidate for the biological control

**Funding:** This work was supported by Korea Institute of Planning and Evaluation for Technology in Food, Agriculture, and Forestry (IPET) through Agricultural Machinery/Equipment Localization Technology Development Program, funded by Ministry of Agriculture, Food, and Rural Affairs (MAFRA) (321054-05-2-HD020). The funders had no role in study design, data collection and analysis, decision to publish, or preparation of the manuscript.

**Competing interests:** The authors have declared that no competing interests exist.

of many agricultural pests, including species of Lepidoptera, Coleoptera, and Diptera [7, 8]. *B. bassiana* is a ubiquitous, mitosporic fungus that is pathogenic on insects and has a broad host range [9]. It is relatively widely used as a microbial control agent for agricultural and forest insect pests. Populations of *B. bassiana* comprise both genetically and phenotypically diverse isolates [9]. Different isolates, which often show wide variation in their virulence to different insects, can easily be obtained from the environment using simple media for their isolation [10]. However, it is well known that entomopathogenic fungi are more virulent on their native host species than on other species [11], although the mortality rate is dependent on not only the fungal strain's host-specificity but also other parameters such as the host-pathogen interactions and the prevailing environmental conditions [12]. The high genetic diversity of *B. bassiana* strains creates potential for microbial control of insect pests and has allowed researchers to continue to develop new mycoinsecticides based on this fungus [13].

Several *B. bassiana* isolates have been assessed and found to be promising biological control agents for coleopteran insects [14–16]. In this study, we report a new strain of *B. bassiana*, designated AAD16, and then we determine the relative pathogenicity of the new isolate (AAD16) against three coleopteran insects, i.e., *Tenebrio molitor* L. (Coleoptera: Tenibrionidae), *Monochamus alternatus* Hope (Coleoptera: Cerambycidae), and *Allomyrina dichotoma* (L.) (Coleoptera: Scarabaeidae), compared to isolate *B. bassiana* ARP14 in a laboratory bioassay. *B. bassiana* ARP14 is effective against different hemipteran and lepidopteran insects [17–19]. The information on the virulence and epizootic conditions of *B. bassiana* that we present here would be helpful in the development of a new EPF strain for use against coleopteran pests.

## Materials and methods

### Source of entomopathogenic fungi

*B. bassiana* AAD16 was isolated from an adult Japanese rhinoceros beetle, *A. dichotoma* collected in Andong, Republic of Korea in 2016 (36.550909, 128.802944). *B. bassiana* ARP14 was recovered from *Riptortus pedestris* (F.) (Hemiptera: Alydidae) in our laboratory. An adult *R. pedestris* infected with *B. bassiana* was collected from a soybean field in Songcheon, Andong, Republic of Korea in 2014 [17].

### Isolation and mass production of the pathogen *B. bassiana*

The infected insect has been preserved in a sterilized falcon tube in a freezer. The fungus was isolated and cultured in Sabouraud Dextrose Agar (SDA) media (Difco™, Maryland, USA) for 14 days. After isolating the fungus from the host, a single colony was removed and cultured, and a pure culture of the *B. bassiana* isolate was produced after two cycles of plating. The purified fungal culture was replated using the loop streak dilution method. A single colony of the fungus was isolated and transferred after 72 hours and then grown for 14 days in SDA media.

### Insect source

Larvae of *T. molitor* were obtained from a commercial insect rearing company (Crazy meal worm, Yongin-si, Republic of Korea) and reared on wheat bran [20]. Larvae were reared at $24.9 \pm 0.0°C$ (degree Celsius), $53.0 \pm 0.9\%$ RH (relative humidity), and a 16:8 hour (Light: Dark) photoperiod in a growth chamber (DS-11BPL, Dasol Scientifc Co. Ltd, Hwaseong, Republic of Korea). Under these rearing conditions, *T. molitor* larvae underwent 12 larval instars (L1-L12).

The Japanese pine sawyer (*M. alternatus*) and Japanese rhinoceros beetle (*A. dichotoma*) were obtained as larvae from a commercial insect rearing company (OsangKinsect Co., Ltd.,

Guri, Republic of Korea). Larvae of *M. alternatus* and *A. dichotoma* were provided with an artificial diet maintained in a growth chamber at 24.8 ± 0.0˚C and 53.5 ± 0.9% RH for *M. alternatus* and 24.9 ± 0.0˚C and 52.3 ± 0.8% RH for *A. dichotoma* with a photoperiod of 16:8 hour (Light:Dark).

## Morphological identification of *B. bassiana* strains

The cultured *B. bassiana* colonies were transferred onto slides with PVA mounting medium (PVA MTNG, BioQuip Products, Gladwick Street, CA, USA) and incubated at 25˚C for 48 hours. The slides were observed under an optical microscope (DM500, Leica, Wetzlar, Germany) with 50× and 100× magnification. The morphology of the fungal pathogen's synnema was studied under scanning electron microscopy (VEGA II LMU, Tescan Orsay Holding, Brno-Kohoutovice, Czech Republic) according to the taxonomic description of Rehner et al. [21].

## Molecular identification of *B. bassiana* strains

Genomic DNA (gDNA) was extracted using a commercial kit (BioFact, Daejeon, Republic of Korea). An internal transcribed spacer (ITS) region was amplified using ITS forward primer (5′ TCCGTAGGTGAACTTGCGG-3′) and ITS reverse primer (5′ TCCTCCGCTTATTGA TATGC-3′) as reported by Schoch et al. [22]. For PCR amplification of the ITS region, the extracted gDNA was used as a template, with 35 cycles under the conditions followed: 1 minute at 94˚C for denaturation, 1 minute at 46˚C for annealing, and 1 minute at 72˚C for extension, and the resulting product was then sequenced by SolGent Co. (Daejeon, Republic of Korea).

The obtained nucleotide sequence of the AAD16 strain (Accession No. MN481507.1) was analyzed using the BlastN program of the National Center for Biotechnology Information (NCBI, www.ncbi.nlm.nih.gov). The evolutionary relationship was inferred using a Neighbor-Joining phylogenetic tree with MEGA6.06 [23]. Bootstrap values on the branches were estimated with 1,000 replications.

## Preparation of conidial suspensions

*B. bassiana* AAD16 and *B. bassiana* ARP14 (Accession No. MG952537.1) were grown under dark conditions at 24.9 ± 0.0˚C and 48.7 ± 0.5% RH for 14 days. Conidial suspensions of the two strains were prepared by scraping the surface of the fungal culture and placing the material obtained into a 20 mL liquid scintillation vial (240804, Wheaton, Millville, NJ) containing autoclaved Triton X-100 (0.1%) solution (Duksan Pure Chemicals Co. Ltd., Ansan, Republic of Korea). To separate the conidial clumps, the suspension was stirred for 2–5 minutes with a Vortex mixer.

Conidial concentrations in the suspensions were measured using Neubauer hemocytometer (Marienfeld-Superior, Paul Marienfeld GmbH and Co. KG, Lauda-KoÈnigshofen, Germany) under a 40× microscope [24]. Based on the count, we adjusted the suspension to a concentration of $1 \times 10^8$ conidia/mL.

## Bioassays

### Topical bioassay on mealworm (*Tenebrio molitor*)

The virulences of the two strains of *B. bassiana* (AAD16 and ARP14) at a concentration of $1 \times 10^8$ conidia/ml were evaluated using two larval sizes (16–18 mm and 22–24 mm) of mealworms (*T. molitor*). Larvae of *T. molitor* were treated with 4 μl of fungal solution on the dorsal

part of abdomen using regular plunger-type syringes and then placed in Petri dishes (60 mm diameter × 15 mm height) and held at 24.7 ± 0.0°C, 99.2 ± 0.1% RH, and 16:8 hour (Light: Dark) photoperiod in the incubator. Triton X-100 ddH$_2$O (0.1%) used as a control. There was one larva in each Petri dish. Bioassays were conducted with 40 larvae in the 16–18 mm size group and 20 larvae in the 22–24 mm size group (1 larva / replication). Mortality of larva was observed at 24 hours intervals from the exposure until death.

### Topical bioassay on long horned beetle (*Monochamus alternatus*)

The virulences of *B. bassiana* strains AAD16 and ARP14 at a concentration of 1×10$^8$ conidia/ml were evaluated on the larvae and adults of *M. alternatus*. Fourteen day-old larvae and seven day-old adults of *M. alternatus* were used in this experiment. Larvae and adults of *M. alternatus* were treated with 4 μl of fungal solution applied on the dorsal part of abdomen using regular plunger-type syringes and then placed in Petri dishes (60 mm diameter × 15mm height) and held at 25.1 ± 0.0°C, 94.4 ± 0.2% RH, and 16:8 hour (Light:Dark) photoperiod in the incubator. Triton X-100 ddH$_2$O (0.1%) was used as a control. In the bioassay, one Petri dish contained a single larva or a single adult. A total of 33 larvae and 8 adults per treatment were used in the bioassays (1 insect / replication). Mortality of larva was observed at 24 hours intervals from the exposure until the death.

### Topical bioassay on Japanese rhinoceros beetle (*Allomyrina dichotoma*)

The virulences of *B. bassiana* strains AAD16 and ARP14 strains at a concentration of 1×10$^8$ conidia/ml were evaluated on larva (larval body size, 7–10 cm) of *A. dichotoma*. Larva of *A. dichotoma* was treated with 50 μl of fungal solution on the dorsal part of abdomen and placed in a breeding dish (100 mm diameter × 40 mm height) and held at 24.9 ± 0.01°C, 96.8 ± 0.3% RH, and 16:8 hour (Light:Dark) photoperiod in the incubator. Triton X-100 ddH$_2$O (0.1%) was used as a control. A total of 20 larvae of *A. dichotoma* were used in the bioassay (1 larva / replication). Mortality of larva was observed at 24 hours intervals from the exposure until 32 days.

### Statistical analysis

Insect mortality was corrected using Abbott's formula [25]. Mortality data for *T. molitor* and *M. alternatus* were subjected to log-probit regression analysis to calculate lethal median time (LT$_{50}$) using SAS 9.4 software [26]. For *A. dichotoma*, survival curves and median lethal times (LT$_{50}$) was estimated using Kaplan-Meier survival analysis because mortality did not reach 100% and compared using the Log-rank test at the 95% level using MedCalc software [27]. Mycosis developmental rates and corrected mortality rates were analyzed using a Chi-square test with a post-hoc multiple comparison test analogous to Tukey's test [28].

## Results

### Morphological identification of *B. bassiana* strains

For strain AAD16, the short, globose-shaped clusters of conidiogenous cells were grouped in ball-shaped structures (Fig 1A and 1B), and the conidia terminated in a rachis with a narrow apical extension (Fig 1C). The zig-zag extension of the elongated rachis formed globose to sub-globose shaped conidia, which are designated as *B. bassiana*.

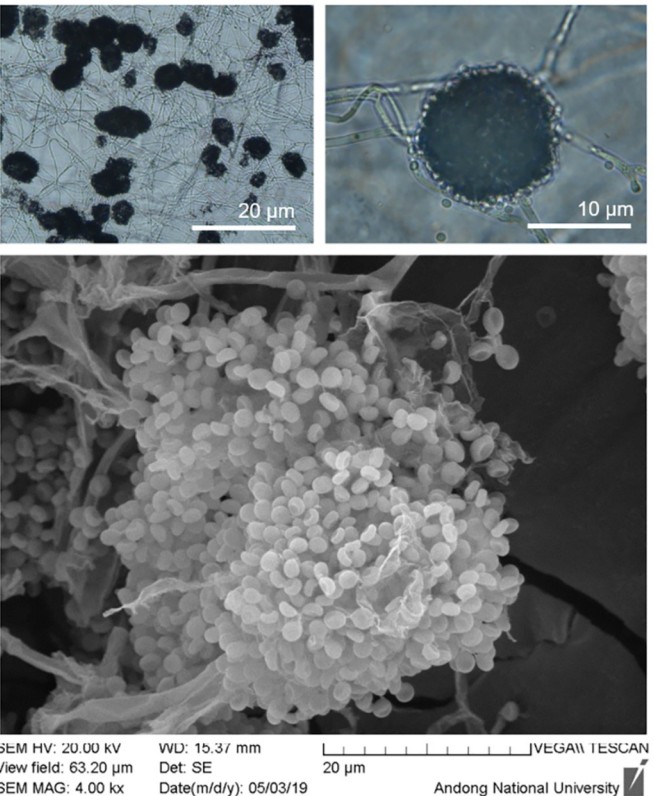

**Fig 1. Scanning electron micrographs of *Beauveria bassiana* AAD16.** (A) Group of clustered conidigenous cells (magnification: 50×), (B) Short globose shaped cluster of conidigenous cells (magnification: 100×), and (C) Conidia shape and rachis structure (magnification: 4.0 K×), arrow indicating the denticulate rachis elongated in a long zig-zag extension; conidia having globose shaped terminated in a narrow apical extension of rachis.

## Molecular identification of *B. bassiana* strains

The ITS region that we amplified had a sequence that showed high similarity (> 99%) with known ITS sequences of several *B. bassiana* strains (Table 1). To evaluate the relationship of the strain with other fungal strains, a phylogenetic tree was constructed using the ITS sequences (Fig 2). This strain AAD16 is designated as *B. bassiana* type clade (Fig 2).

## Topical bioassay on mealworm (*Tenebrio molitor*) larvae

Significant differences in mealworm mortality were found between the two fungal strains tested within each of the two mealworm larval sizes (16–18 mm and 22–24 mm) (Fig 3). *B.*

**Table 1. Identification of entomopathogenic fungal strain using ITS sequence (GenBank accession No. MN481507.1).**

| Species Blasted in NCBI-GenBank | GenBank Accession Number | Total Score | E Value | Identity (%) |
|---|---|---|---|---|
| *B. Bassiana* isolate BbI6 | KX263272.1 | 924 | 0 | 99.16 |
| *B. Bassiana* isolate JEF006 | KT280276.1 | 923 | 0 | 99.81 |
| *B. Bassiana* isolate WGS11782 | JX406519 | 923 | 0 | 99.81 |
| *B. Bassiana* isolate V1 | KR139926.1 | 922 | 0 | 99.61 |
| *B. Bassiana* isolate BbN22A02 | Mk952565.1 | 921 | 0 | 99.81 |

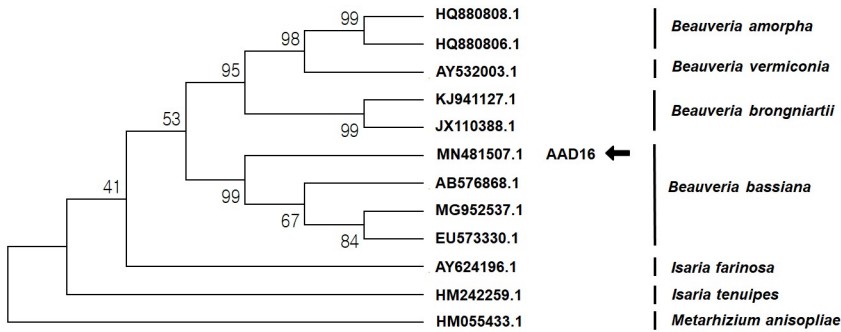

**Fig 2. Molecular phylogenetic analysis of *Beauveria bassiana* AAD16 with similar strain and other genera based on nucleotide sequence and constructed by maximum likelihood method.**

*bassiana* strain AAD16 caused higher mortality in both sizes of mealworms compared to the *B. bassiana* strain ARP14.

The $LT_{50}$ value of *B. bassiana* AAD16 was 115.9 hours ($\chi^2 = 3.98$, $df = 7$, $P = 0.782$), which was significantly lower than for *B. bassiana* ARP14, i.e., $LT_{50}$ value of 136.9 hours ($\chi^2 = 8.34$, $df = 14$, $P = 0.871$) on larvae in the 16–18 mm size group (Table 2). Strain AAD16 also caused the highest corrected mortality at three time points after exposure, namely 80.0% ($Z = 2.38$, $df = 1$, $P = 0.017$), 95.0% ($Z = 2.36$, $df = 1$, $P = 0.018$), and 100.0% ($Z = 3.65$, $df = 1$, $P < 0.001$ at 6,8 and 19 days, respectively, after exposure, as compared to *B. bassiana* ARP14 (Fig 3).

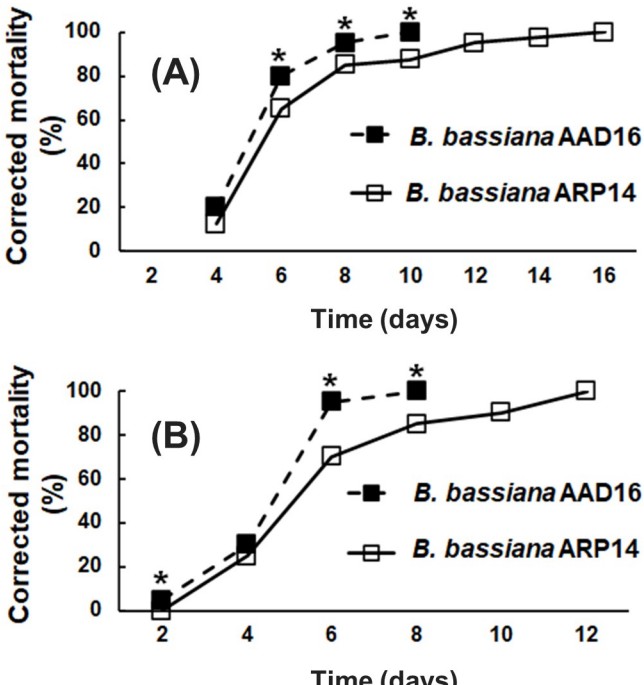

**Fig 3. Efficacy of *Beauveria bassiana* AAD16 and ARP14 strains on mealworm (*Tenebrio molitor*) larvae in the size of (A) 16-18 and (B) 22–24 mm over time.** The * above bars indicate significant differences among the treatments (Tukey studentized range HSD test, $\alpha = 0.05$).

**Table 2. Calculation of LT$_{50}$ values for two larval sizes of mealworm larvae of *Tenebrio molitor*.**

| Body length of larvae tested (mm) | n | Fungus strain tested | LT$_{50}$ (hours) | 95% CI | Slope ± SE | $\chi^2$ (df) |
|---|---|---|---|---|---|---|
| 16–18 | 40 | *B. bassiana* AAD16 | 115.87 a | 108.95–122.59 | 7.87 ± 0.75 | 3.98 (7) |
| | 40 | *B. bassiana* ARP14 | 136.87 b | 127.63–145.68 | 5.61 ± 0.40 | 8.34 (14) |
| | 40 | Control[&] | - | - | - | - |
| 22–24 | 20 | *B. bassiana* AAD16 | 99.65 a | 89.92–109.62 | 7.00 ± 1.06 | 7.86 (5) |
| | 20 | *B. bassiana* ARP14 | 123.69 b | 110.88–135.77 | 5.40 ± 0.62 | 2.85 (10) |
| | 20 | Control[&] | - | - | - | - |

[&]No mortality occurred in control

LT$_{50}$ value followed by the same letters is not significantly different among the treatment based on 95% C.I.

The rate of mycosis was also significantly different between the two fungal strains 10 days after exposure ($\chi^2$ = 98.43, *df* = 2, *P* < 0.001) (Fig 4); at 10 days post-exposure, the mycosis rate was 100.0% in *B. bassiana* AAD16 and 85% for strain ARP14 (Fig 4).

For the larger sized mealworm larvae (22–24 mm), the LT$_{50}$ value caused by strain AAD16 was 99.7 hours ($\chi^2$ = 7.86, *df* = 5, *P* = 0.164), which was significantly shorter than for strain ARP14, which had an LT$_{50}$ value of 123.7 ($\chi^2$ = 2.86, *df* = 10, *P* = 0.985) (Table 2). At 2, 6, and 8 days after exposure, the AAD16 strain caused higher corrected mortality than did the other strain: 2 day = 5.0% (*Z* = 2.26, *df* = 1, *P* = 0.024), 6 day = 95.0% (*Z* = 4.65, *df* = 1, *P* < 0.001), and 8 day = 100.0% (*Z* = 4.03, *df* = 1, *P* < 0.001) (Fig 3). The mycosis rate was 100.0% in *B. bassiana* AAD16 which was significantly higher than *B. bassiana* ARP14 (80.0%) on 8 days of exposure (Fig 4).

## Topical bioassay on long horned beetle (*Monochamus alternatus*)

The virulence of *B. bassiana* AAD16 strain against first instar larvae (14 days old) and adults of *M. alternatus* was significantly higher than that for *B. bassiana* ARP14 strain. The LT$_{50}$ value

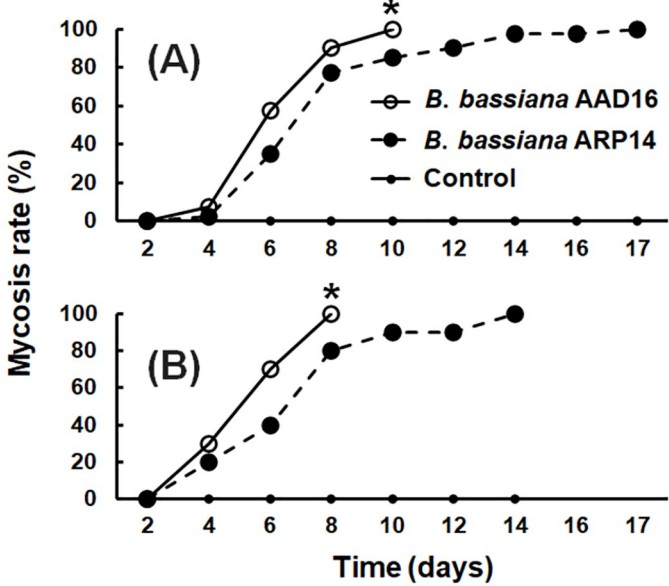

**Fig 4. Mycosis rate (%) of *Beauveria bassiana* AAD16 and ARP14 strains on mealworm (*T. molitor*) larvae in the size of (A) 16–18 and (B) 22–24 mm.** The * above the data point in line graph indicate significant differences among the treatments (Tukey studentized range HSD test, $\alpha$ = 0.05).

**Table 3. Calculation of LT$_{50}$ values for 14-d-old long horned beetle (*Monochamus alternatus*).**

| Life stages | Fungus strain tested | n | LT$_{50}$ (hours) | 95% CI | Slope ± SE | $\chi^2$ (df) |
|---|---|---|---|---|---|---|
| Larvae | *B. bassiana* AAD16 | 33 | 75.84 a | 70.82–80.35 | 13.91 ± 2.44 | 0.35 (7) |
| | *B. bassiana* ARP14 | 33 | 98.79 b | 91.95–105.34 | 8.64 ± 1.06 | 2.33 (9) |
| | Control$^{\&}$ | 33 | - | - | - | - |
| Adults | *B. bassiana* AAD16 | 8 | 180.62 a | 155.46–205.04 | 5.53 ± 0.99 | 3.65 (12) |
| | Control | 6 | 341.54 b | 306.03–398.44 | 5.79 ± 1.18 | 1.30 (16) |

$^{\&}$No mortality occurred in control

LT$_{50}$ value followed by the same letters is not significantly different among the treatment based on 95% C.I.

of *B. bassiana* AAD16 on larvae was 75.8 hours ($\chi^2$ = 0.35, *df* = 7, *P* = 0.999) which was significantly lower than that caused by *B. bassiana* ARP14 ($\chi^2$ = 2.33, *df* = 9, *P* = 0.985) (Table 3). The lethal median time (LT$_{50}$) of *B. bassiana* AAD16 (18.6 hours) for adults was also lower than for the control (341.5 hours) (Table 3). The corrected mortality of *M. alternatus* larvae was 100.0% when treated with *B. bassiana* AAD16 strain, which was significantly higher than the ones obtained with *B. bassiana* ARP14 strain (68.8%) after 120 hours of exposure (*Z* = 6.09, *df* = 1, *P* < 0.001) (Fig 5). The mycosis rate was 97.0% in *B. bassiana* AAD16, which was significantly higher than *B. bassiana* ARP14 (54.4%) after 120 hours of exposure ($\chi^2$ = 62.39, *df* = 2, *P* < 0.001) (Fig 6).

## Topical bioassay on Japanese rhinoceros beetle (*Allomyrina dichotoma*)

The virulence of *B. bassiana* AAD16 strain was also higher on the larvae of *A. dichotoma* compared to *B. bassiana* ARP14, with a LT$_{50}$ of 17.7 days for AAD16 (Table 4). However, the corresponding lethal median time (LT$_{50}$) could not be obtained for *B. bassiana* ARP14 because mortality lower than 50% did not occur after 30 days. Survival analysis *A. dichotoma* exposed to entomopathogenic fungi showed significant differences among treatments ($\chi^2$ = 16.36, *df* = 2, *P* < 0.001). After 30 days of exposure, the survival rate of *A. dichotoma* larvae treated with *B. bassiana* AAD16 was 5.0%, whereas for larvae treated with the *B. bassiana* ARP14 strain, the survival rate was 55.0% (Fig 7).

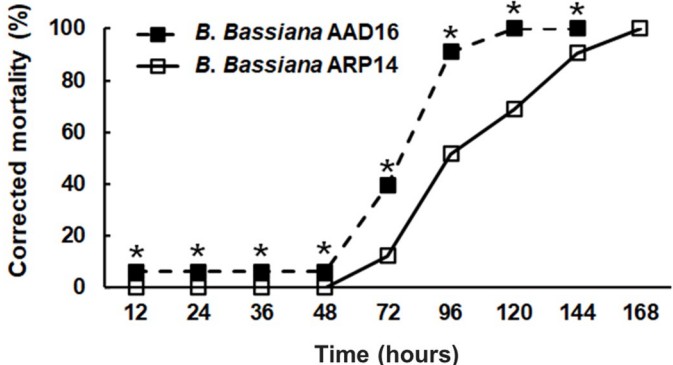

**Fig 5. Virulence of *Beauveria bassiana* AAD16 and ARP14 strains on 14 days old long horned beetle (*Monochamus alternatus*) larvae.** The * above bars indicate significant differences among the treatments (Tukey studentized range HSD test, $\alpha$ = 0.05).

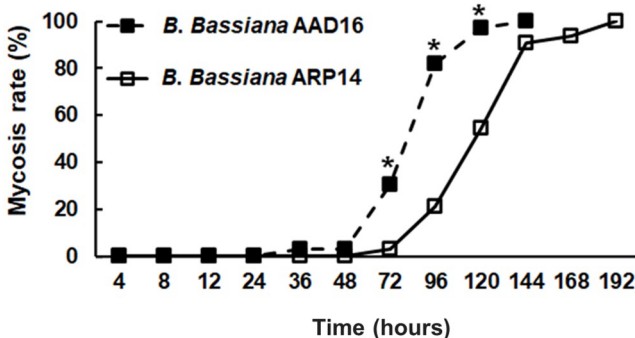

**Fig 6. Mycosis rate of *Beauveria bassiana* AAD16 and ARP14 strains on 14 days old long horned beetle (*M. alternatus*) larvae.** The * above the data point in line graph indicate significant differences among the treatments (Tukey studentized range HSD test, $\alpha = 0.05$).

## Discussion

The new entomopathogenic fungal isolate collected from *A. dichotoma* was identified as *B. bassiana* and designated as strain AAD16 based on morphology [29] and intraspecies and interspecies divergence rate with different *Beauveria* species and strains [30]. Entomopathogenic fungi are important pathogens of many arthropod pests such as aphids, leafhoppers, whiteflies, stink bugs, lepidopterans, and coleopterans [31–38]. The efficacy of the entomopathogenic fungus varies from strain to strain [39], so we evaluated two strains of *B. bassiana* (ARP14 and AAD16) against three coleopteran insects, i.e., *T. molitor*, *M. alternatus*, and *A. dichotoma*.

*Tenebrio molitor* is known for its susceptibility to entomopathogenic fungal infection and is a suitable test organism for assessing fungal virulence [40]. Strain AAD16 was more virulence than the ARP14 strain to *T. molitor* larvae when applied topically. In the study by Rodríguez-Gómez et al. [41] who reported similar virulence levels for two *B. bassiana* isolates, i.e., the wild type and its mutant type, against the larvae of *T. molitor*, the $LT_{50}$ value of mutant type isolate was 5.7 days, which was longer than $LT_{50}$ value of 4.2 days that we found in our study for large larvae (22–24 mm). In another study by Praprotnik et al. [40], among five *B. bassiana* isolates, *B. bassiana* 2121 showed higher virulence to larvae of *T. molitor*.

For the longhorned beetle *M. alternatus*, the *B. bassiana* AAD16 strain killed the host faster, causing 100% mortality by day 5 after exposure while our other treatment, *B. bassiana* ARP14, caused only 69% mortality in the same time period. Sone et al. [42] reported that exposing adult *M. alternatus* to non-woven fabric strips containing *B. bassiana* spores at $1.4 \times 10^{8}$ conidia/cm$^2$ caused 97% mortality in 14 days. Other strains of *B. bassiana*, such as F-263 and ERL836, have also been shown to be effective against *M. alternatus* [15, 43, 44]. In another

**Table 4. Calculation of $LT_{50}$ values for *A. dichotoma*.**

| Fungus strain tested | $LT_{50}$ (days) | 95% CI |
|---|---|---|
| *Beauveria bassiana* AAD16 | 17.70 | 14.50–20.90 |
| *Beauveria bassiana* ARP14[&] | - | - |
| Control[&] | - | - |

[&]Mortality did not occur above 50%

$LT_{50}$ value followed by the same letters is not significantly different among the treatment based on 95% C.I.

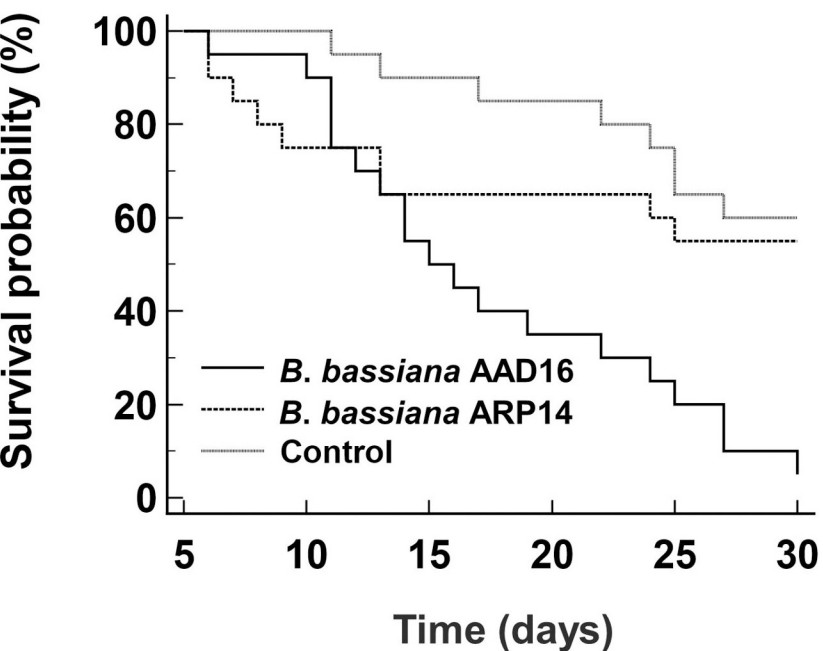

**Fig 7. Survival curves of *Allomyrina dichotoma* exposed *Beauveria bassiana* AAD16 and ARP14.**

study [45], application of *B. bassiana* F-263 at $5.5\times10^6$ conidia/individual of on 10 day-old beetles killed 50% of *M. alternatus*, while younger beetles that were only 4 days old, suffered 50% mortality from a lower dose ($1.9\times10^6$ conidia/individual) in a somewhat longer time period (by 14 days) when conidial mixtures were applied to the tarsi of $CO_2$- anesthetized adults with a fine hairbrush. However, in our experiment where higher concentration of fungus was applied, *B. bassiana* AAD16 kills beetles faster, causing 50% adult mortality in this species within only 8 days.

For our third test species (*A. dichotoma*, the beetle from which the AAB16 strain was recovered), we found that the virulence of the AAD16 strain was higher than that for the other strain ARP14. *Allomyrina dichotoma* larvae exposed to *B. bassiana* AAD16 had a low (5.0%) survival rate after 30 days of exposure, compared to a substantial (55.0%) survival rate at the same time point if treated with the other strain, *B. bassiana* ARP14. Nevertheless, *B. bassiana* ARP14 is highly virulent against *R. pedestris*, the host from which it was originally isolated [17]. Similarly, another *B. bassiana* isolate (F-263), collected from a cadaver of a larva of *M. alternatus*, was most virulent against its original host [43, 46]. In Japan and China, isolates of *B. bassiana* have been successfully applied for the control of *M. alternatus* using inoculative methods of fungal bands and baited traps, respectively [47] and so, having the most virulent strain available is important in achieving good control with these programs against *M. alternatus*.

The basis for the above discussed intra-specific variation in virulence in this group of pathogens could be any of several modes of action [35] known in these fungi, including mechanical damage resulting from tissue invasion, depletion of nutrient resources and toxicosis, and production of toxins in host insect body [48]. The toxins Beauvericin, Bassianolide, Isarolides, and Beauverolides have been isolated from *B. bassiana* infected hosts [49, 50], but the capacity to produce toxic compounds varies depending on the strain of *B. bassiana* [39].

The mortality rate of insect hosts caused by entomopathogenic fungus could vary depending on the fungal strain, host-pathogen interaction, fungal specificity for the host, and other

factors [12]. Strains are known to often be more virulent on their natal host species than on novel species [11]. As *B. bassiana* AAD16 strain was isolated from *A. dichotoma* while *B. bassiana* ARP14 strain from *R. pedestris*, *B. bassiana* AAD16 showed higher virulence to the three coleopteran insects we tested than did ARP14. This finding may indicate that *B. bassiana* AAD16 is probably more specific to coleopteran insects. Thus, we conclude that *B. bassiana* AAD16 has the potential to be used as microbial control agent against coleopteran insects.

## Supporting information

**S1 File. Raw data *Tenebrio molitor*, raw data_*Monochamus alternatus*, raw data_*Allomyrina dichotoma*.**
(XLSX)

## Author Contributions

**Conceptualization:** Un Taek Lim.

**Data curation:** Hyong Woo Choi, Un Taek Lim.

**Formal analysis:** Souvic Sarker.

**Funding acquisition:** Un Taek Lim.

**Investigation:** Souvic Sarker.

**Methodology:** Souvic Sarker, Hyong Woo Choi.

**Project administration:** Un Taek Lim.

**Resources:** Un Taek Lim.

**Supervision:** Un Taek Lim.

**Writing – original draft:** Souvic Sarker.

**Writing – review & editing:** Souvic Sarker, Hyong Woo Choi, Un Taek Lim.

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
