## [Decision Letter · Decision Letter 0]

16 Oct 2023

PONE-D-23-23339Evaluation of new strain (AAD16) of Beauveria bassiana recovered from Japanese rhinoceros beetle: effects on three coleopteran insectsPLOS ONE

Dear Dr. Lim,

Thank you for submitting your manuscript to PLOS ONE. After careful consideration, we feel that it has merit but does not fully meet PLOS ONE’s publication criteria as it currently stands. Therefore, we invite you to submit a revised version of the manuscript that addresses the points raised during the review process.

We look forward to receiving your revised manuscript.

Kind regards,

Rachid Bouharroud

Academic Editor

PLOS ONE

https://academic.oup.com/jinsectscience/article/22/2/2/6550598?login=false

https://www.j3.jstage.jst.go.jp/article/aez/42/4/42_4_563/_pdf

https://journals.plos.org/plosone/article?id=10.1371%2Fjournal.pone.0195848

In your revision ensure you cite all your sources (including your own works), and quote or rephrase any duplicated text outside the methods section. Further consideration is dependent on these concerns being addressed.

“This work was supported by Korea Institute of Planning and Evaluation for Technology in Food, Agriculture, and Forestry (IPET) through Agricultural Machinery/Equipment Localization Technology Development Program, funded by Ministry of Agriculture, Food, and Rural Affairs (MAFRA) (321054-05-2-HD020).”

 “This work was supported by Korea Institute of Planning and Evaluation for Technology in Food, Agriculture, and Forestry (IPET) through Agricultural Machinery/Equipment Localization Technology Development Program, funded by Ministry of Agriculture, Food, and Rural Affairs (MAFRA) (321054-05-2-HD020).”

6. We note that Figure 1 in your submission contain copyrighted images. All PLOS content is published under the Creative Commons Attribution License (CC BY 4.0), which means that the manuscript, images, and Supporting Information files will be freely available online, and any third party is permitted to access, download, copy, distribute, and use these materials in any way, even commercially, with proper attribution. For more information, see our copyright guidelines: http://journals.plos.org/plosone/s/licenses-and-copyright.

Additional Editor Comments:

Dear Author

Really it was not easy for me to take a decision about your paper since there was 2 reviewers that recommend your paper to be published after "minor revision" and the 3rd one rejected the paper. That why i read it carefully trice.

Based on that, I ask you to address also the comments of the reviewer 2. For nevelty, TEF molecular charachterization and non practical use of syringue at field please forgot it but for the following comments you should give a feedback:

1- Mycosis observation on died insects ;

2- Was the fungal strain re-isolated from the insect that undergone mycosis?

3- On which larval stages of insects, was the bioassay done?

Of course the comments of the reviewers 1 and 3 should be addressed.

Regards

Reviewers' comments:

Reviewer's Responses to Questions

**Comments to the Author**

1. Is the manuscript technically sound, and do the data support the conclusions?

Reviewer #1: Yes

Reviewer #2: Partly

Reviewer #3: Yes

2. Has the statistical analysis been performed appropriately and rigorously? 

Reviewer #1: N/A

Reviewer #2: Yes

Reviewer #3: Yes

3. Have the authors made all data underlying the findings in their manuscript fully available?

Reviewer #1: Yes

Reviewer #2: No

Reviewer #3: Yes

4. Is the manuscript presented in an intelligible fashion and written in standard English?

Reviewer #1: Yes

Reviewer #2: Yes

Reviewer #3: No

5. Review Comments to the Author

Reviewer #1: General Comments: (Minor Comments)

The manuscript titled " Evaluation of new strain (AAD16) of Beauveria bassiana recovered from Japanese rhinoceros beetle: effects on three coleopteran insects " explores the potential of new Beauveria species as biological control agents against the three coleopteran insects. The study addresses a relevant research area and offers important insights for pest management strategies using biological averages. This study confirms a broad spectrum of Bauveria species as entomopathogenic fungi.

The manuscript is generally well structured, and the experiments are carefully designed. However, a few points require improvement before the manuscript can be considered suitable for publication in PLOS ONE.

• Line 4: replace “….its virulency” with “its virulence”,

• Line 5: change “on three coleopteran..” by “focusing on its effect on three coleopteran”,

• Line 7: 15.3 !! After comma, two number is usually used to convey precision without overwhelming the reader. try to apply this in all documents!

• Line 7 : replace “ and 19,4% lower on strain AAD16 » by and 19,4% lower for strain AAD16

• Line 9 : replace “1.3 and 1.2 times higher than strain ARP14” by “1.3 and 1.2 times higher than that of strain ARP14”

• Line 11 : replace “47.1% lower on strain “ by “47.1% lower for strain”

• Line 13 : replace “1.8 higher than strain” by “1.8 higher than that of strain”

• Line 14 : replace ‘’ the strain AAD16 also showed higher mortality (90.0%) for larvae of A. dichotoma than did the strain ARP14’’ by ‘’the strain AAD16 also showed higher larval mortality (90%) for A. dichotoma compared to strain ARP14’’

• Line 25 : The Latin names of the species must be written entirely the first time in the text (ex: Beauveria bassiana (line 25). After the following repetitions (line 30, 48 ….) they can be abbreviated on B. bassiana, and this applies equally to all other species mentioned in the text.

• Line 60 : added the company of the Sabouroud Dextrose Agar

• Line 61 : added the conditions of incubation ( T° , RH , photoperiod)

• Line 64 : We isolated a single colony of the fungus and transferred after 72 h… into what (medium ) ??

• Line 69 : replace “53± 0.9%” by “53 ± 0.9 % RH “

• Line 66 : replace “Insect Source” by “Insect source”

• Line 76 : replace “and 24.9±0.00°C and 52.3± 0.8% RH for with a photoperiod of 16 :8h (L :D)’’ by “and 24.9±0.00°C and 52.3± 0.8% RH for A. dichtoma with a photoperiod of 16 :8h (L :D)

• line 79 : replace ‘’ Morphological Identification of B. bassiana strains ‘’ by Morphological “identification of B. bassiana strains”

• line 115 : in this paragraph its better to mention the formula used to calculate the mortality of larva

• line 119 : replace “petri dishes” by “Petri dishes”

• line 119 : did you used the sterilized Petri dishes or non-sterilized ?

• line 120 : homogenize the writing of photoperiod conditions in the all of document, 16 :8( L : D)

• line 124 : in this paragraph, it's better to mention the formula used to calculate the larva and adult mortality of M. alternatus

• line 127 : replace “were tested” by “were used”

• line 129 : replace “petri” by “Petri”

• line 166 : replace Beauveria bassiana by B . bassiana

• line 174 : replace “at 6,8 and 19 days after exposure” by “at 6,8 and 19 days, respectively, after exposure”

• Figures and tables are generally well-presented and support the findings. However, it is advisable to include more descriptive captions (statistic information (P-value)) that succinctly explain the content

• Figure legends 3-6 : the absence of standard error bars of the mean and asterisk's meaning.

• Reference: line 245: Citation number 45 is missing in the text. Please ensure that all references are cited according to the required journal style.

The manuscript could provide valuable information in the field of pest management. However, major revisions are needed (minor comments mentioned above). It is recommended that the authors revise the manuscript accordingly and provide additional details in the methodology section to improve the rigor of the study and its reproducibility for future researchers. Thank you for the opportunity to review this work. I look forward to seeing the revised manuscript and the contributions it will make to the field.

Sincerely,

Reviewer #2: This research work presented minimum amount of data and lacks of novelty. Only one Beauveria bassiana isolate was isolated and characterized and compared its efficacy with an in-house isolate against larvae of three insect through a single bioassay test. For molecular characterization, ITS region was used, but using more specific region like TEF would be good for this type pleomorphic fungus. However, my main concern is that the method of fungal inoculation by using a syringe is inappropriate and practically impossible in field use. The virulence of entomopathogenic fungi includes the invasion capabilities through insect cuticle through germination on host body. This manuscript did not show any visual evidence of mycosed larvae which is essential. In addition, the following question need to be addressed.

1. Was the fungal strain re-isolated from the insect that undergone mycosis?

2. On which larval stages of insects, was the bioassay done?

3. What is the effect of this strain on other larval stages? Is there any sub-lethal effect?

Reviewer #3: The author reported “Evaluation of new strain (AAD16) of Beauveria bassiana recovered from Japanese rhinoceros beetle: effects on three coleopteran insects” which is interesting because the entomopathogenic fungus Beauveria bassiana can be considered as a potential biological control agent against coleopteran pests. But this article needs to revise some critical points that are listed in the bellow comments.

Comments

1. Rewrite the objective of your study

2. Line 42 from “in this study” to line 50 “insects” should be in Materials and methods section

3. In each experiment, how much larvae and/or adults were used

4. Indicate how much repetition in each experiment

5. Homogenate 16: 8 (L:D)

6. The h, d, D, L, dia and RH should be clear.

7. The English need a revision by a mother tongue to improve the language.

8. L22 change including fungus…by including fungi

9. L32 but also on other parameters

10. L56 Isolation and mass production of the pathogen Beauveria bassiana

11. L69 0.9% RH

12. L75 delete provided by the company

13. L 92 under the conditions followed: under the following conditions

14. L119 Petri dishes

15. L131 and L 138 Triton X-100 ddH2O (0.1%) used as a control

16. L194 change “ for the” by “ the ones obtained with”

17. L225 we found that the B. bassiana AAD16 strain killed

18. L227 B. bassiana ARP14, caused only only 69% mortality

6. PLOS authors have the option to publish the peer review history of their article (what does this mean?). If published, this will include your full peer review and any attached files.

Reviewer #1: No

Reviewer #2: No

Reviewer #3: No

---

## [Author Response · Author response to Decision Letter 0]

30 Nov 2023

*** 1. Response to editor’s comments

1. Mycosis observation on died insects

>> Yes. We observed the mycosis development on dead insects. We addressed this issue in our previous MS in figure 4 and 6.

2. Was the fungal strain re-isolated from the insect that undergone mycosis?

>> Yes, for confirmation purposes, we only re-isolated the fungus from T. molitor and checked for molecular identification using the ITS primer which confirmed our targeted organism.

3. On which larval stages of insects, was the bioassay done?

>> It was very difficult to separate the larval stages due to long larval period, thus overlapped larval stages. Thus, we provided the larval sizes for T. molitor and A. dichotoma and the larval age for M. alternatus (see L121, L132, and L144 in the revised MS).

*** 2. Response to reviewer #1’s comments

>> We found mismatch in line numbers reviewer indicated, thus we provided the line numbers of the first MS in parenthesis with red font next to the line numbers reviewer indicated.

1. The manuscript titled " Evaluation of new strain (AAD16) of Beauveria bassiana recovered from Japanese rhinoceros beetle: effects on three coleopteran insects " explores the potential of new Beauveria species as biological control agents against the three coleopteran insects. The study addresses a relevant research area and offers important insights for pest management strategies using biological averages. This study confirms a broad spectrum of Beauveria species as entomopathogenic fungi.

>> Thanks for the comments.

2. Line 4: replace “….its virulency” with “its virulence”,

>> Replaced “its virulency” to “its virulence” (see L04 in the revised MS).

3. Line 5: change “on three coleopteran..” by “focusing on its effect on three coleopteran”,

>> Replaced “on three coleopteran” to “focusing on its effect on three coleopteran” (see L05-06 in the revised MS).

4. Line 7: 15.3 !! After comma, two number is usually used to convey precision without overwhelming the reader. try to apply this in all documents!

>> We set three significant figures for the data.

5. Line 7: replace “ and 19,4% lower on strain AAD16 ? by and 19,4% lower for strain AAD16

>> Replaced “and 19.4% lower on strain AAD16” to “and 19.4% lower for strain AAD16” (see L07-08 in the revised MS).

6. Line 9: replace “1.3 and 1.2 times higher than strain ARP14” by “1.3 and 1.2 times higher than that of strain ARP14”

>> Replaced “1.3 and 1.2 times higher than strain ARP14” to “1.3 and 1.2 times higher than that of strain ARP14” (see L09-10 in the revised MS).

7. Line 11 : replace “47.1% lower on strain “ by “47.1% lower for strain”

>> Replaced “47.1% lower on strain” to “47.1% lower for strain” (see L12 in the revised MS).

8. Line 13 : replace “1.8 higher than strain” by “1.8 higher than that of strain”

>> Replaced “1.8 higher than strain” to “1.8 higher than that of strain” (see L13 in the revised MS).

9. Line 14: replace ‘’ the strain AAD16 also showed higher mortality (90.0%) for larvae of A. dichotoma than did the strain ARP14’’ by ‘’the strain AAD16 also showed higher larval mortality (90%) for A. dichotoma compared to strain ARP14’’

>> Replaced “The strain AAD16 also showed higher mortality (90.0%) for larvae of A. dichotoma than did the strain ARP14” to “The strain AAD16 also showed higher larval mortality (90%) for A. dichotoma compared to strain ARP14’” (see L14-15 in the revised MS).

10. Line 25: The Latin names of the species must be written entirely the first time in the text (ex: Beauveria bassiana (line 25). After the following repetitions (line 30, 48 ….) they can be abbreviated on B. bassiana, and this applies equally to all other species mentioned in the text.

>> Replaced “Beauveria bassiana” to “B. bassiana” (see L30, 106, and 176 in the revised MS).

11. Line 60: added the company of the Sabouroud Dextrose Agar

>> Added the company name in the revised MS (see L63 in the revised MS).

12. Line 61: added the conditions of incubation ( T° , RH , photoperiod)

>> We addressed this issue in our previous MS in L103-104 (see L106-107 in the revised MS).

13. Line 64: We isolated a single colony of the fungus and transferred after 72 h… into what (medium ) ??

>> Added “in SDA media” (see L67 in the revised MS).

14. Line 69: replace “53± 0.9%” by “53 ± 0.9 % RH “

>> Replaced “53± 0.9%” to “53.0 ± 0.9% RH” (see L72 in the revised MS).

15. Line 66: replace “Insect Source” by “Insect source”

>> Replaced “Insect Source” to “Insect source” (see L69 in the revised MS).

16. Line 76: replace “and 24.9±0.00°C and 52.3± 0.8% RH for with a photoperiod of 16 :8h (L :D)’’ by “and 24.9±0.00°C and 52.3± 0.8% RH for A. dichtoma with a photoperiod of 16 :8h (L :D)

>> Replaced “and 24.9±0.0°C and 52.3± 0.8% RH for with a photoperiod of 16 :8h (L :D)” to “and 24.9 ± 0.0°C and 52.3 ± 0.8% RH for A. dichotoma with a photoperiod of 16:8 hour (Light:Dark)” (see L79-80 in the revised MS).

17. line 79: replace ‘’ Morphological Identification of B. bassiana strains ‘’ by Morphological “identification of B. bassiana strains”

>> Replaced “Morphological Identification of B. bassiana strains” to “Morphological identification of B. bassiana strains” (see L82 in the revised MS).

18. line 115: in this paragraph its better to mention the formula used to calculate the mortality of larva

>> We believe it is OK as we provided reference (see L152 in the revised MS).

19. line 119 (L120): replace “petri dishes” by “Petri dishes”

>> Replaced “petri dishes” to “Petri dishes” (see L123 in the revised MS).

20. line 119 (L120): did you used the sterilized Petri dishes or non-sterilized ?

>> We did not sterilize the Petri dishes by ourselves, we just used brand-new (pre-sterilized) Petri dishes in all bioassays.

21. line 120 (L121): homogenize the writing of photoperiod conditions in the all of document, 16 :8( L : D)

>> Changed “16 L: 8 D” to “16:8 hour (Light:Dark)” (see L72, L124-125, 136, and L147 in the revised MS).

22. line 124: in this paragraph, it's better to mention the formula used to calculate the larva and adult mortality of M. alternatus

>> We believe it is OK as we provided reference (see L152 in the revised MS).

23. line 127 (L128): replace “were tested” by “were used”

>> Replaced “were tested” to “were used” (see L133 in the revised MS).

24. line 129 (L130): replace “petri” by “Petri”

>> Replaced “petri” to “Petri” (see L135 in the revised MS).

25. line 166 (L167): replace Beauveria bassiana by B . bassiana

>> Replaced “Beauveria bassiana” to “B. bassiana” (see L176 in the revised MS).

26. line 174: replace “at 6,8 and 19 days after exposure” by “at 6,8 and 19 days, respectively, after exposure”

>> Replaced “at 6, 8 and 19 days after exposure” to “at 6, 8, and 19 days, respectively, after exposure” (see L183-184 in the revised MS).

27. Figures and tables are generally well-presented and support the findings. However, it is advisable to include more descriptive captions (statistic information (P-value)) that succinctly explain the content

>> Statistic information with P-values were added in figure captions (see L440-441, L443-445, L447-448, L450-451, and L453 in the revised MS).

28. Figure legends 3-6: the absence of standard error bars of the mean and asterisk's meaning.

>> We did the Chi-square test using the total number of insects; that’s why there are no standard errors. We explain the meaning of the asterisk sign in the figure caption (see L440-441, L443-445, L447-448, and L450-451 in the revised MS).

29. Reference: line 245: Citation number 45 is missing in the text. Please ensure that all references are cited according to the required journal style.

>> The citation number 45 was presented in the first MS in L232 (see L241 in the revised MS). 

*** 3. Response to reviewer #2’s comments

1. This research work presented minimum amount of data and lacks of novelty. Only one Beauveria bassiana isolate was isolated and characterized and compared its efficacy with an in-house isolate against larvae of three insect through a single bioassay test. For molecular characterization, ITS region was used, but using more specific region like TEF would be good for this type pleomorphic fungus. However, my main concern is that the method of fungal inoculation by using a syringe is inappropriate and practically impossible in field use. The virulence of entomopathogenic fungi includes the invasion capabilities through insect cuticle through germination on host body. This manuscript did not show any visual evidence of mycosed larvae which is essential. In addition, the following question need to be addressed.

>> We agree with the reviewer’s comment. Firstly, we compared the virulence of our new fungus strain (AAD16) with another B. bassiana strain (ARP14) because, in our previous study, we found B. bassiana ARP14 is more virulent than the commercial strain B. bassiana GHA on hemipteran and lepidopteran insects. We mentioned this issue in the first MS in L49-51.

Secondly, yes, using a syringe is inappropriate and practically impossible in field use. As our intention was to compare the efficacy of two fungus strains, we wanted to apply the same amount of conidia from both fungi to the insect body using a microsyringe.

2. Was the fungal strain re-isolated from the insect that undergone mycosis?

>> Yes, for confirmation purposes, we only re-isolated the fungus from T. molitor and checked for molecular identification using the ITS primer which confirmed our targeted organism.

3. On which larval stages of insects, was the bioassay done?

>> It was very difficult to separate the larval stages due to long larval period, thus overlapped larval stages. Thus, we provided the larval sizes for T. molitor and A. dichotoma and the larval age for M. alternatus (see L121, L132, and L144 in the revised MS).

4. What is the effect of this strain on other larval stages? Is there any sub-lethal effect?

>> We did not test larvae other than described in the text nor evaluate the sublethal effect. 

*** 4. Response to reviewer #3’s comments

>> We found mismatch in line numbers reviewer indicated, thus we provided the line numbers of the first MS in parenthesis with red font next to the line numbers reviewer indicated.

1. Rewrite the objective of your study

>> We rewrite the objective as " In this study, we report a new strain of B. bassiana, designated AAD16, and then we determine the relative pathogenicity of the new isolate (AAD16) against three coleopteran insects, i.e., Tenebrio molitor L. (Coleoptera: Tenibrionidae), Monochamus alternatus Hope (Coleoptera: Cerambycidae), and Allomyrina dichotoma (L.) (Coleoptera: Scarabaeidae), compared to isolate B. bassiana ARP14 in a laboratory bioassay. B. bassiana ARP14 is effective against different hemipteran and lepidopteran insects [17-19]” (see L43-48 in the revised MS).

2. Line 42 from “in this study” to line 50 “insects” should be in Materials and methods section

>> Deleted L43-51 from the previous MS and made a new section in Materials and Methods in the revised MS (see L54-59 in the revised MS). We also deleted L58-59, “An adult A. dichotoma infected with B. bassiana was collected from field in Andong, Republic of Korea in 2016 (36.550909, 128.802944),” in the first submission.

3. In each experiment, how much larvae and/or adults were used

>> Added larval or adult number (see L125-127, L137-139, and L148 in the revised MS).

4. Indicate how much repetition in each experiment

>> Added number of replication (see L125-127, L137-139, and L148 in the revised MS).

5. Homogenate 16: 8 (L:D)

>> Changed “16 L: 8 D” to “16:8 h (Light:Dark)” (see L72, L124-125, 136, and L147 in the revised MS).

6. The h, d, D, L, dia and RH should be clear.

>> h, d, D, L, dia and RH changed to hour, day, Dark, Light, diameter, and relative humidity, respectively ( see L9, 13, 15, 64, 67, 72, 80, 85, 107, 124, 128, 135, 136, 139, 146, 147, 149, 179, 180, 185, 189, 191, 194, 198, 200, 202, 203, 205, 207, 212, 214, 216, 233, 244, 248, 252, Table 2-4, 446, and 449 in the revised MS).

7. The English need a revision by a mother tongue to improve the language.

>> We made the first MS revised by a commercial company (www.vandrieschescientificediting.com) as we indicated in the first submission. If this revised MS still needs English revision, we will submit it to the company again. Please let us know. 

8. L22: change including fungus…by including fungi

>> Changed “fungus” to “fungi” (see L22 in the revised MS).

9. L32: but also on other parameters

>> We couldn’t understand this comment. Please rephrase it in next revision.

10. L56: Isolation and mass production of the pathogen Beauveria bassiana

>> Changed “Isolation and mass production of the pathogen” to “Isolation and mass production of the pathogen B. bassiana” (see L61 in the revised MS).

11. L69: 0.9% RH

>> Added “RH (relative humidity)” (see L72 in the revise MS).

12. L75: delete provided by the company

>> Deleted “provided by the company” (see L78 in the revised MS).

13. L 92: under the conditions followed: under the following conditions

>> Changed “under the following conditions” to “under the conditions followed” (see L95-96 in the revised MS).

14. L119: (L120): Petri dishes

>> Changed “petri dishes” to “Petri dishes” (see L123 in the revised MS).

15. L131 and L 138 (L139): Triton X-100 ddH2O (0.1%) used as a control

>> We corrected as “Triton X-100 ddH2O (0.1%) was used as a control” (see L136-137 and L147-148 in the revised MS).

16. L194 (L195): change “ for the” by “ the ones obtained with”

>> Changed “for the” to “the ones obtained with” (see L204-205 in the revised MS).

17. L225 (L226): we found that the B. bassiana AAD16 strain killed

>> We deleted “we found that” (see L236 in the revised MS).

18. L227 B. bassiana ARP14, caused only only 69% mortality

>> Deleted “only” (see L238 in the revised MS).

 

*** 5. Authors’ self-corrections (Line numbers are from the first MS)

L62: Deleted space (see L65 in the revised MS)

L69: Added “(degree Celsius)” (see L72 in the revised MS).

L93,108: Changed “min” to “minute” (see L96 and L111 in the revised MS).

L121, 130: Changed “h” to “height” (see L124 and L135 in the revised MS).

L178, 182: Deleted space (see L187 and L191 in the revised MS).

L184: Added space (see L193 in the revised MS).

L258: Deleted space (see L268 in the revised MS).

L275-277: We added “Supporting information”

Table 2: Deleted “(n= 20)” from the caption and added a column for “n” 

Figure 3, 7: Changed “d” to “days” (see Figure 3 and Figure 7 in the revised MS).

Figure 5, 6: Changed “h” to “hours” (see Figure 5 and Figure 6 in the revised MS).

---

## [Editor Report · Decision Letter 1]

6 Dec 2023

Evaluation of new strain (AAD16) of Beauveria bassiana recovered from Japanese rhinoceros beetle: effects on three coleopteran insects

PONE-D-23-23339R1

Dear Dr. Lim,

We’re pleased to inform you that your manuscript has been judged scientifically suitable for publication and will be formally accepted for publication once it meets all outstanding technical requirements.

Kind regards,

Rachid Bouharroud

Academic Editor

PLOS ONE

Additional Editor Comments (optional):

Dear

Your responses to reviewers regarding your manuscript (PONE-D-23-23339) are currently convincing. The 3 comments i have underlined and based mainly on reviewer 2 comments were adressed. Also i appreciate self-corrections parts to improve you paper.

Good luck

Regards
---

## [Editor Report · Acceptance letter]

18 Dec 2023

PONE-D-23-23339R1 

PLOS ONE

Dear Dr. Lim, 

I'm pleased to inform you that your manuscript has been deemed suitable for publication in PLOS ONE. Congratulations! Your manuscript is now being handed over to our production team.

Kind regards, 

on behalf of

Dr. Rachid Bouharroud 

Academic Editor

PLOS ONE